# The Effect of Cropping Systems on the Dispersal of Mycotoxigenic Fungi by Insects in Pre-Harvest Maize in Kenya

**DOI:** 10.3390/insects15120995

**Published:** 2024-12-16

**Authors:** Ginson M. Riungu, James Muthomi, Maina Wagacha, Wolfgang Buechs, Esther S. Philip, Torsten Meiners

**Affiliations:** 1Sugar Research Institute, Kenya Agricultural and Livestock Research Organization, Kisumu P.O. Box 44-40100, Kenya; 2Department of Plant Science and Crop Protection, Faculty of Agriculture, University of Nairobi, Nairobi P.O. Box 29053-0625, Kenya; 3Department of Biology, Faculty of Science and Technology, University of Nairobi, Nairobi P.O. Box 30197-00100, Kenya; 4Institute for Biology and Chemistry, University of Hildesheim, Universitaetsplatz 1, 31141 Hildesheim, Germany; 5Kenya Plant Health Inspectorate Service, Nairobi P.O. Box 49592-00100, Kenya; 6Julius-Kuehn Institute, Koenigin-Luise-Str. 19, 14195 Berlin, Germany

**Keywords:** push–pull, *Trichoderma*, *Aspergillus*, *Fusarium verticillioides*, aflatoxins, maize–legume intercropping, zoochory

## Abstract

Ongoing climate change has led to increased insect damage to maize and the mycotoxigenic fungal infestation and subsequent mycotoxin contamination of maize meant for food and feed. A field study was conducted in two regions in Kenya to see how different maize–legume cropping systems affect the arthropod taxa that were most prevalent on maize at the flowering and grain-filling stages, the arthropods that were most damaging, those that could potentially disperse mycotoxigenic fungi on pre-harvest maize, and the aflatoxin contamination of the grain. Our work revealed that the main herbivore in maize is the fall armyworm (FAW), which was prevalent in both regions but was significantly diminished by the push–pull cropping system. The presence of *Aspergillus* and *Fusarium verticillioides* on the exoskeleton of maize weevils, sap beetles, earwigs, and carpenter ants suggests a potential passive dispersal of the fungi in pre-harvest maize. The fungi have previously been isolated from maize from the two regions of Kenya. They are associated with the production of aflatoxins and fumonisins, which present a serious hazard to human and animal health. To reduce maize contamination with mycotoxigenic fungi, farmers can apply targeted insect management strategies, including intercropping and push–pull technology.

## 1. Introduction

The impact of different maize cropping systems on pests and diseases in maize may vary, potentially influencing yield and food safety in different ways. Maize–legume intercropping, a common practice among smallholder farmers in East Africa, holds the potential to significantly boost land and labour utilization [1]. The positive outcomes of intercropping maize with leguminous crops such as common bean (*Phaseolus vulgaris*), mung bean (*Vigna radiata*), fava bean (*Vicia faba*), and soya bean (*Glycine max*) are well documented and include enhanced soil fertility, reduced disease occurrence, and increased overall productivity [2,3,4]. This promising approach offers a ray of hope for the future of crop management and food safety.

Maize–legume intercropping may significantly improve the diversity of beneficial entomofauna in smallholder agricultural production systems [5]. Intercropping increases the population of beneficial insects and decreases the population of certain insect pests, such as the budworm (*Heliothis* spp.), the corn borer (*Ostrinia nubilalis*), the leafhopper (*Cicadulina mbila*), and the maize stalk borer (*Busseola fusca*) [6,7,8]. It is reported that insects more easily locate hosts, feed more, and have higher reproduction rates in monocrops than in intercrops [9]. Intercrop plants close to insect targets may interfere with herbivores’ cognition and localization of the host [10]. In addition, intercrops affect some host plants’ quality and chemical suitability [11]. This enlightening aspect of intercrop systems underscores their potential in sustainable crop management.

The push–pull technique is an innovative agricultural method widely used in Africa to enhance food security and sustainable farming practices. This approach involves integrating the use of specific plant varieties that repel pests (push) and those that attract beneficial insects (pull). Farmers can reduce pest damage, improve soil health, and increase yields by strategically planting these crops together [12,13]. Farmers in western Kenya have embraced push–pull technology, which uses Napier grass, *Pennisetum purpureum* Schumach., as the ‘pull’ and *Desmodium* spp. as the ‘push’. This technology has significantly reduced aflatoxin contamination in maize [14,15]. Maize intercropping with *Desmodium* (a repellent crop) and surrounding fields with Napier grass (an attractive trap crop) have been shown to reduce maize crop damage by Lepidopteran pests [12,13].

*Trichoderma harzianum* is a beneficial fungus widely recognized for its role in combating soil-borne mycotoxigenic fungi, including *Aspergillus*. This biocontrol agent works by outcompeting harmful fungi for resources, thereby inhibiting their growth and reducing aflatoxin production. Additionally, *T. harzianum* enhances soil health and promotes plant growth, making it a valuable tool in integrated pest management strategies [16]. *Trichoderma harzianum* has been shown to parasitize *Aspergillus flavus* as well as colonize the fungal entry points [17]. Additionally, *T. harzianum* biodegrades aflatoxin B1 in maize grains [18]. In Kenya, the use of *T. harzianum* is often combined with maize–legume intercropping.

*Aspergillus flavus* infection in maize and its potential contamination with aflatoxin, which poses a significant health risk to humans and animals, are challenges of significant concern. The susceptibility of maize to *A. flavus* infection is influenced by various factors, including insect infestation, grain damage, and environmental conditions [19]. Insect damage coupled with favourable climatic conditions like high temperatures and drought stress usually results in enhanced aflatoxin contamination [20]. The European corn borer, *Ostrinia nubilalis* (Hubner); the corn earworm, *Helicoverpa zea* (Boddie); and the FAW, *Spodoptera frugiperda* (J.E. Smith), have been identified as significant contributors to *A. flavus* infection and subsequent aflatoxin contamination in pre-harvest maize [19]. Climate change has led to extended drought stress and high temperatures, conditions that favour *Aspergillus* infection and subsequent aflatoxin contamination in maize [21]. In addition, climate change increases the geographic range and population densities of insect pests [22].

While previous studies have explored the benefits of maize–legume intercrops in terms of productivity per unit area, soil fertility improvement, soil conservation, and related economic benefits [5,23,24], our study focused on the effect of maize–legume intercrops on the occurrence of insect pests, population densities, and their role in enhancing *Aspergillus* and aflatoxin contamination.

The objective of this study was to test the hypothesis that intercropping maize with legumes, *T. harzianum* application, and the push–pull method will reduce damage to maize by herbivores and reduce the dispersal of mycotoxigenic fungi and associated aflatoxin contamination, while maintaining or increasing productivity. This study had two main objectives: firstly, to evaluate the insects that are potential vectors for *Aspergillus* in pre-harvest maize, and secondly, to determine the mechanisms of mycotoxigenic fungi dispersal. The results of this study could inform the development of more sustainable pest management strategies and contribute to a reduction in the aflatoxin contamination in maize.

## 2. Materials and Methods

### 2.1. Description of the Study Sites and Trial Establishment

This study was conducted at the Kenya Agricultural and Livestock Research Organization (KALRO) farms in Kibos, Kisumu County (0°2′11″ S, 34°49′17″ E), and Kambi ya Mawe, Makueni County (01°37′ S, 37°40′ E), as shown in Figure 1. The treatments were as follows: (1) maize monocrop, (2) maize–bean intercrop, (3) maize–bean intercrop sprayed with *Trichoderma harzianum* T22 (Koppert, Kenya), and (4) push–pull technique (maize intercropped with *Desmodium intortum* and with three rows of Napier grass (*Pennisetum purpureum*). The *Desmodium* and Napier grass were pre-germinated for planting to ensure survival in Makueni, which receives lower amounts of rain. *Trichoderma harzianum* was mixed in water at a 1:5 product-to-water ratio, as per the manufacturer’s instructions, and sprayed on the maize planting holes. The intercrop crops were planted at a ratio of one row of legume to two rows of maize on the dates and with the spacing shown in Table 1. The land was prepared using a tractor-mounted disc harrow, and the trial was established using a randomized complete block design (RCBD) with three replicates. The plots were 30 m long and 30 m wide to minimize the effect between the treatments. The plots were randomly allocated as per the standard layout shown in Appendix A, with a 2-metre buffer between the plots. Two seeds were placed per hole and thinned to one plant per spot after germination. The varieties planted, the spacing, and the rainfall data are shown in Table 1. In Kisumu, planting was performed on 6 April 2021 and 11 October 2021 for the long and short rain cropping seasons, respectively, while in Makueni, it was performed on 3 April 21 and 23 November 2021, respectively. Di-ammonium phosphate fertilizer (18:46:0, NPK) was used at planting at a rate of 125 kg/ha and was top-dressed 30 days after planting with Calcium Ammonium Nitrate fertilizer with 21% nitrogen at 200 kg/ha. Weeding was performed by hand and no pesticides were used in the trial so as not to affect the arthropods.

### 2.2. Collection, Identification, and Enumeration of Arthropods

Insects were captured fortnightly during the generative stage of maize (BBCH 69–-89) [25] between 800 h and 1200 h to ensure comparable results. Insects found on the maize ears or silks of 20 pre-tagged plants were handpicked or drawn into sterile vials using a pooter. Only plants at the middle of the plot (the 5 innermost rows) were tagged to minimize effects from other treatments, particularly from *Desmodium*. The insects were singly placed in 1.5 mL centrifuge tubes and labelled with plot numbers and dates for further analysis.

The arthropods were identified at the lowest possible taxonomic level using morphological characters under a stereo dissecting microscope (Wild M38, Leica, Heerbrugg, Switzerland) with the help of keys and catalogues [26,27], and confirmation was performed by Mr. Morris Mutua, an entomologist at the National Museums of Kenya. A list of the arthropods sampled from the plots was made, with each taxon represented through means per treatment [28]

The FAW larvae were noted as the most destructive herbivores damaging both foliage and cobs across the two regions, and since records of effective management using the push–pull method have been reported, data on the incidence and severity of attacks were determined for 20 plants per plot that were randomly selected from the 5 innermost rows. The incidence on foliage or ears was determined to be the percentage of plants or ears showing an attack by the larvae, respectively, while the damage severity was estimated using a scale (1 = low to 5 = high) [29].

### 2.3. Aspergillus Isolation from Insects Captured on Maize Ears

The insects were processed as described by Yamoah et al. [30] and Awad et al. [31], with modifications in the number of individuals per species and the replicates. Twenty individuals from each species caught per plot, per collection event, were washed off separately to dislodge fungi from each insect’s exoskeleton. Each beetle was placed in a sterile universal bottle containing 3 mL of 100 mmol Potassium phosphate buffer (pH 7.0) + 0.01% Tween 80 and shaken for three minutes in a vortex machine (Vortex-Genie 2, Scientific Industries, New York City, NY, USA). The washed insect samples were serially diluted 100-fold by successively pipetting 1 mL of the sample into a sterile tube and topping it up with 9 mL of sterile distilled water. A 1 ml aliquot of each dilution series (0, 10^−1^, and 10^−2^) was plated on Petri dishes containing potato dextrose agar (PDA) with chloramphenicol (39 g of PDA, Oxoid Ltd, Basingstoke, UK, and 250 mg of chloramphenicol). The plates were incubated at room temperature (25 ± 2 °C) and with a 12 h photoperiod for five days, after which the colony-forming units (CFUs) were counted and the population was expressed as CFUs per insect.

### 2.4. Detection of Viable Aspergillus and Fusarium Spores on Beetles in Pre-Harvest Maize

A slightly different isolation technique [32] with modifications was used to determine the mode of spread (zoochory or endozoochory) of *Aspergillus* and *Fusarium* spores. Twenty *Sitophilus zeamais*, *Carpophilus* spp., and *Forficula* spp. individuals, each captured as described above in Makueni County from maize at the BBCH 75, 85, and 87 developmental stages, were put into pre-sterilized Petri dishes. Insects from Makueni had a higher number of *Aspergillus* spores in the initial tests described above, so only insects from Makueni were considered. The insects were incubated for 36 h at 25 ± 2 °C, with a 12:12 h L–D photoperiod, and at 72 ± 10% RH on laboratory benches. The insects were shaded with a paperboard and supplied with water using a slightly moist sterile cotton bud. After 36 h, the insects were put into a refrigerator at 4 °C for 1 h to slow down their metabolic activity. Fecal pellets dropped in the Petri dish during the 36 h incubation period were picked aseptically using a sterile scalpel and placed on PDA with chloramphenicol. The head, the elytra, and the guts were aseptically detached from the insects using sterile forceps in a laminar flow with the aid of a stereo dissecting microscope. The head and elytra samples were cultivated directly onto media as described above, whereas the gut was surface-sterilized in 70% ethanol for 10 s, rinsed in sterile distilled water, and punctured before plating, as described above.

The fungi were incubated for 5 days at 25 ± 2 °C. Following this incubation period, colonies morphologically identified as *Aspergillus* [33] or *Fusarium* [34] were enumerated and subcultured on PDA for confirmation purposes. The calmodulin and elongation factor gene markers were used to identify the *Aspergillus* and *Fusarium* isolates, respectively [35].

### 2.5. Maize and Legume Harvesting, Sample Handling, and Analysis

Maize was harvested manually at physiological maturity (BBCH 89) [22]. Five primary ears from the 20 pre-tagged plants per plot were systematically (every 4th plant) harvested and dehusked in order to evaluate the extent of ear rot based on kernel discolouration. This was conducted through a visual assessment of grain colour and development, with scores ranging from 1 (indicating no damage or discolouration) to 5 (indicating severe damage or discolouration) [36]. The second batch was subjected to a manual shelling process, followed by a sun-drying procedure (using a Twist Grain Pro device manufactured in Draminski, Poland) until the moisture content was reduced to below 13%. Subsequently, the kernels underwent a fine milling process using a coffee and spice grinder (AR1100, Moulinex, Ecully, France). The blender was cleaned and rinsed between samples with 70% ethanol to prevent cross-contamination. Grain yield was quantified by multiplying the average grain yield of the ten pre-tagged plants in each plot by the number of plants in one hectare (44,000 plants ha^−1^). The weight was determined using an analytical balance (Nimbus 1602E, Adam Equipment, Milton Keynes, UK). The percentage of spoiled grain was determined by counting the spoiled grain from a random sample of 100 kernels in a bag in four replicates.

The 100-seed weight was determined by averaging the weight of the four replicates of 100 seeds used to determine the maize spoilage. The weights were measured using an analytical balance (Nimbus 1602E, Adam Equipment, UK). The bean and *Desmodium* yields were determined by averaging the bean grain and *Desmodium* forage harvested from four replicates of randomly chosen 1 m^2^ areas per plot and extrapolated by multiplying by 10,000 m^2^ (the size of a hectare of land).

### 2.6. Total Aflatoxin Content Determination

The total aflatoxin content of 10 g of flour was determined using the total aflatoxin assay (Helica, Biosystems Inc., Santa Ana, CA, USA). This assay is based on a solid-phase competitive inhibition enzyme immunoassay with an aflatoxin-specific antibody optimized to cross-react with all four subtypes of aflatoxin (B1, B2, G1, and G2) in grain [37].

Aflatoxin extraction was conducted using 70% methanol (300 mL of de-ionized water was added to 700 mL of methanol) as the extraction solvent. Five grams of milled maize flour was added to 25 mL of the extraction solvent using a 1.5 (weight by volume) (*w*/*v*) ratio. The mixture was agitated in an orbital shaker for a period of two minutes, after which it was left to stand for a further two minutes to allow any particulate matter to settle. Ten millilitres of the supernatant was filtered into a clean beaker using Whatman #1 filter paper [37].

For the assay, aliquots of 100 µL of the sample or of standard solution, in duplicates, were added to a mixing well with 200 µL of the aflatoxin–HRP conjugate and mixed by priming the pipettor thrice. From the mix, 100 µL of the solution was pipetted into corresponding wells in an antibody-coated microtitre well and incubated at room temperature for 15 min. The contents of the wells were then discarded, and the microwells were washed off five times by filling each of the wells with a phosphate-buffered saline–Tween (PBS–Tween) buffer. The microtitre plates were dried by inverting them on absorbent paper towels. A total of 100 µL of substrate reagent was added to each well. The plates were incubated in a dark chamber for 5 min to avoid direct light, and the reaction was stopped by adding 120 µL of the stop solution to each well. Each microwell’s optical density (O.D.) readings (Eliza Reader, ELx 808, Biotek, Houston, TX, USA) at 450 nm, were noted. A standard curve was constructed using the mean relative absorbance of the standard references against their concentrations in ng/mL on a logarithmic curve. Mean sample relative absorbance values were extrapolated to the corresponding concentrations.

The formula for relative absorbance is as follows:$$\%\;\mathrm{Relative}\;\mathrm{absorbance}=\frac{Absorbance\;standard}{Absorbance\;zero\;standard}\times 100;$$

### 2.7. Data Analysis

Data collected were subjected to SAS version 9 for an analysis of variance (ANOVA) at *p* ≤ 0.05. The mean ± SE number of arthropods per plant in the specific cropping system and maize development stages (BBCH) was calculated. Data on maize yield, grain spoilage, kernel weight, bean yields, aflatoxin levels, and fungal colonization were subjected to ANOVA. Because the grain yield, % spoiled grain, CFU/g, and aflatoxin levels (ppb) were not normally distributed, they were log-transformed (log_10_). Post-hoc tests were performed using Tukey’s honestly significant difference (Tukey’s HSD) procedure at a *p* ≤ 0.05 level of significance for each trait determined whenever the main effects were significant.

## 3. Results

### 3.1. FAW Incidence and Damage on Maize Foliage and Cobs

The FAWs were identified as the most damaging insects in maize. The FAW larvae attacked the maize foliage and later moved into the cobs as the crop matured (Figure 2) (Table 2). The percentage of foliage exhibiting damage was found to be significantly influenced by location (F = 29.4, df = 1, *p* < 0.001), season (F = 15.2, df = 1, *p* < 0.001), and treatment (F = 29.4, df = 3, *p* < 0.001). The severity differed between the two locations (F = 132.2, df = 1, *p* < 0.001) and among treatments (F = 42.7, df = 3, *p* < 0.001). The highest incidence of damage was in Makueni during the long rain cropping season, at 85.8%, and the least was in Kisumu during the long rain cropping season, at 45.7%.

Damage to maize foliage and the incidence on foliage and cobs were highest in the maize monocrop treatment, significantly differing from those in the maize–legume intercropping systems. The FAW incidence on foliage was highest, at 75%, in maize monocrops and lowest (41%) in the push–pull cropping system. A similar trend was observed concerning the severity of damage in foliage and the percentage of incidence on cobs. In the long and short rain seasons, the highest FAW incidence and severity were recorded in Makueni. Incidences of 100% were recorded on the cobs and foliage, particularly during the long rain cropping season. In Kisumu, the damage by the FAW was lower than that in Makueni, and the cobs were not substantially attacked.

### 3.2. Recovery of Microorganisms from Insects Captured from Maize

Among the arthropods, the most frequently observed taxa, in descending order, were *Forficula* spp., *Helicoverpa zea*, *Spodoptera frugiperda*, *Carpophilus* spp., *Sitophilus zeamais*, and *Aplomya* sp., among others (Figure 3). Four insect taxa, namely, *Sitophilus zeamais*, *Carpophilus* spp., *Forficula* spp., and *Camponotus* spp., were identified from the list of insects analyzed for fungal spore load on their exoskeletons (Table 3). These four taxa exhibited a significant number of mycotoxigenic fungi spores on their bodies. The predominant fungal genera isolated from the insects captured were *Aspergillus* and *Fusarium*. Other fungi genera detected included *Penicillium* and *Diplopodia* at insignificant levels. The site of collection significantly influenced the *Aspergillus* spore load on *S. zeamais* (*p* = 0.004), *Carpophilus* spp. (*p* = 0.009), and *Camponotus* spp. (*p* = 0.034), and the *Fusarium* spore load on *Forficula* spp. (*p* = 0.008). The season influenced the *Aspergillus* spore load on *S. zeamais* (*p* = 0.004), *Carpophilus* spp. (*p* = 0.015), and *Forficula* spp. (*p* = 0.009), and the *Fusarium* spore load on *S. zeamais* (*p* = 0.035) and *Forficula* spp. (*p* = 0.007). The maize weevil (*S. zeamais*) harboured the highest *Aspergillus* spore load (125.8), whereas the sugar ants (*Camponotus* spp.) had the lowest no. of *Aspergillus* spores (5.0) on their exoskeletons. Similarly, *S. zeamais* harboured a very high *Fusarium* spore load (176.1) on their exoskeletons, while the *Carpophilus* spp. had the lowest CFUs (11.4) on their exoskeletons. The site and the season significantly influenced the recovery of *Aspergillus* and *Fusarium* from the insects’ exoskeletons. *Aspergillus* load was highest in Makueni during the long rain cropping season, whereas *Fusarium* was higher during the short rain season than during the long rain cropping season. The main *Aspergillus* species isolated were *A. flavus*, *A. minisclerotigenes*, *A. japonicus*, and *A. niger*, whereas all the *Fusarium* specimens were identified as *Fusarium verticillioides*.

### 3.3. Mechanism of Fungal Spore Spread by Coleopterans

The highest prevalence of *Aspergillus* infestation was observed in *S. zeamais* specimens (52.5%), followed by *Carpophilus dimidiatus* (27.1%) and *Forficula* spp. (26%). In contrast, *C. dimidiatus* was more infested with *F. verticilloides* (35%), followed by *S. zeamais* and *Forficula* sp., at 29.5% and 23.2%, respectively (Table 4). In both fungal species, the elytra exhibited the greatest prevalence of spores, followed by the head, gut, and feces, in descending order.

### 3.4. Maize and Companion Crop Yield Parameters

The cropping system significantly influenced the percentage of grain spoilage (F = 6.65, df = 3, *p* < 0.001) and the aflatoxin levels in maize kernels (F = 8.97, df = 3, *p* < 0.001). Maize yield within a site did not differ significantly. However, the yields were significantly different between seasons (F = 10.55, df = 1, *p* = 0.03) and from location to location (F = 3.49, df = 1, <0.001). The highest maize yields (in kg/ha) were observed in the maize monoculture, while the lowest yields were recorded in the maize–bean intercrop (Table 5). The intercrop yields were found to be comparable to one another. The short rain season yielded a higher number of crops than the long rain season, and the yields in Kisumu were three times higher than in Makueni. *Desmodium* yields were four times higher in Kisumu than in Makueni.

Kernel spoilage differed significantly between sites (F = 3.10, df = 1, <0.001) and cropping systems (F = 6.64, df = 3, *p* = 0.045). The highest grain spoilage was observed in the maize monocultures, while the lowest was observed in the push–pull cropping system. The highest kernel spoilage was recorded in Makueni, while the lowest recorded came from Kisumu. The 100-seed weight was found to be highest during the long rain season, differing significantly (F = 15.3, df = 1, *p* < 0.001) from that in the short rain season. The highest levels of aflatoxin contamination were observed in Makueni during the long rain season and in the maize monocrop, with levels influenced by several factors including site, season, and cropping system. Bean yields were highest in Kisumu and during the long rain season. The cropping system did not influence the bean yields.

## 4. Discussion

The ongoing effects of climate change have resulted in increased insect damage to maize crops, as well as the proliferation of mycotoxigenic fungal infestations, which have subsequently led to the contamination of maize intended for human consumption and animal feed with mycotoxins [12,15]. Here, a field study in two regions of Kenya investigated the impact of different maize–legume cropping systems on the arthropod taxa most prevalent on maize at the flowering and grain-filling stages, on the arthropods that cause the most damage, on those that could potentially disperse mycotoxigenic fungi on pre-harvest maize, and on the aflatoxin contamination of the grain.

### 4.1. FAW Damage

The *Desmodium* in push–pull technology systems significantly reduced the abundance of *Spodoptera frugiperda* pest insects and crop damage. It is known that *Desmodium* reduces Lepidopteran pests when intercropped with cereals [5,12,38]. However, the mechanism of the management strategy is still under debate. It is not clear whether *Desmodium* repels the pests or intercepts and kills them. Intercropping can interrupt the visual orientation of pests to their hosts. It can also interrupt olfactory host-finding mechanisms with volatile chemical compounds [39].

### 4.2. Recovery of Microbials from Insects Captured from Maize

*Aspergillus* and *Fusarium* were the mycotoxigenic fungi recovered from the insects trapped in the two regions studied. Although arthropod spore dispersal in pre-harvest maize has not been studied in Kenya before, studies of the fungal contamination of maize in farms, markets, and farm stores have been reported [40,41,42]. These studies isolated *Aspergillus*, *Fusarium*, and *Penicillium* at varying levels. Thus, the isolation of similar fungal species in and on beetles captured from the same areas in the field reveals the risk posed to humans and animals that rely on maize for food and feed. *Aspergillus* and *Fusarium* are harmful pathogens of maize that produce secondary metabolites and toxins under favourable conditions. Toxigenic species and strains of the two fungi isolated from the insects are potential producers of toxic secondary metabolites (aflatoxins and fumonisins) [43,44]. In this study, maize kernels had mean total aflatoxin levels of 4.9 and 1.9 ppb in the Makueni and Kisumu counties, respectively. Although this is below the maximum allowable limit of 10 ppb, it still poses the threat of chronic aflatoxicosis [45,46].

#### Mode of Dispersal of Aspergillus and Fusarium Spores in Pre-Harvest Maize

In the present study, many insects carried viable *Aspergillus* and *Fusarium* spores on their elytra and on their heads. In contrast, only a few of the insects had viable spores in the gut or the feces. The high number of spores on the exoskeleton suggests that the dispersal of mycotoxigenic fungi is primarily passive, which is in agreement with the findings of [47,48], who, although studying fungal dispersal in stored grain, concluded that weevils played a role in the dispersal of *Aspergillus* and that dispersal was primarily passive. A greenhouse experiment in Kenya [49] showed that both *S. zeamais* and *C. dimidiatus* increased *A. flavus* and aflatoxin contamination in pre-harvest maize. Among the sap beetles, many individuals had viable spores in the gut and the feces. The high number of spores in the guts of sap beetles may be due to possible faunivory or the accidental ingestion of fungal propagules when feeding on plant material (endozoochory) [50]. Many species of sap beetles in the Nitidulidae taxon are herbivores or fungivores that are attracted to damaged maize plants or plants with exposed kernels. There, they feed on fungi that develop on the exudates from plant wounds or directly on the kernels [51]. Zoochory in maize has been studied in relation to ear rot by *F. verticillioides* [52], and the authors reported that the rootworm enhanced ear rot. However, they did not investigate the mechanisms of interaction.

### 4.3. Maize and Companion Crop Yields

Maize yields were higher in the short rain season than in the long rain season. The average yields for both locations were 4.9 and 5.8 tons per ha in the short and long rain seasons, respectively. The difference in yield is attributed to the climatic conditions during the trial periods. The levels of rain during the long rain cropping season were high (over 20 mm per day on some days), particularly in Kisumu, and could have caused a reduction in yields due to flooding and soil leaching. Although rain is generally good for maize growth, too much rainfall can cause nitrogen to leach out of nutrient-poor soils, leading to negative feedback and lower yields [53].

Kisumu had an average maize yield of 8 tons per ha, compared to 2.7 tons in Makueni. The considerable variance in yield is attributed to the difference in climatic conditions between the two regions. Kisumu usually receives more favourable rainfall than Makueni. During the trial period, Kisumu received 1714 mm of rainfall, compared to 828 mm in Makueni. According to [54], estimates of climate change show a trend towards lower maize yields in some locations, with temperature increases above certain thresholds contributing to severe yield losses. Rainfall accounts for 44% of the variance in maize yields [55].

In terms of cropping systems, maize monocultures produced the highest yields. Higher yields in monocultures may be attributed to the lack of intraspecific competition for resources. This finding is in line with Pierre et al. [5], who indicated that the yield advantages of monocultures over maize–bean intercropping are due to interspecific competition between cereal–legume species for nutrients, space, water, and light. The competition for resources between maize and the intercrops may result in decreased yields of maize [56]. The maize–legume intercrops were comparable in terms of maize yield. Although their maize yields were lower than that in the monocrops, the yields of the companion crops (beans and fodder) would supplement the farmers’ total yield. However, bean and *Desmodium* yields behaved similarly to the maize yields, suggesting that they were equally affected by the weather and the agro-ecological sites in the same way as maize. Beans are the third most important crop and a source of dietary protein [57]. The fact that the intercropping did not significantly reduce the maize yields therefore represents a benefit to farmers, who can easily meet their dietary requirements by adding a protein source without compromising their maize yields. Similarly, by using push–pull technology, the farmer can easily obtain fodder for his cows while preserving his food source.

### 4.4. Grain Spoilage, Fungal Infestation, and Mycotoxin Concentration

Grain spoilage was higher in Makueni than in Kisumu. The higher grain spoilage in Makueni can be attributed to the high level of FAW damage. In Makueni, the FAW damage was so severe that the incidence was 100% in both leaves and cobs. Cob damage was lowest in the push–pull treatment, with the lowest grain spoilage. Similar results regarding the efficacy of push–pull cropping in reducing the damage caused by FAWs to maize by up to 50% have been reported [15,58]. The present study shows that push–pull technology effectively reduced FAW damage and subsequent aflatoxin contamination in maize. Although the FAW larvae did not carry *Aspergillus* spores, we hypothesize that the heavy damage to the ears in Makueni, coupled with the drought situation and the availability of the aflatoxigenic *Aspergillus* species, enhanced the aflatoxin levels in maize.

Grain spoilage was lower in maize–bean intercrops and in maize–bean intercrops with *Trichoderma*. The reduced damage in the intercrops echoed the findings of [59], which, studying the effect of maize–bean intercropping, reported that the intercrops had lower FAW infestation than the maize monocrops. Aflatoxin contamination was also lower in kernels from maize–legume intercrops than in those from maize monocrops due to less severe damage caused by the herbivorous lepidopterans and less subsequent infestation by *Aspergillus* species.

Among the maize from the counties, maize from Makueni had higher aflatoxin contamination than that from Kisumu. This can be attributed to the high prevalence of *Aspergillus* species that are known to synthesize aflatoxins in Makueni [34,35] and to drought stress on maize at flowering.

## 5. Conclusions and Recommendations

In the present study, maize–legume intercrops and push–pull technology enhanced general insect abundance. At the same time, the intercrops reduced pest damage to maize crops, resulting in a decline in aflatoxin contamination in maize. Although the maize yield was lower in the intercrops, the bean grain yield in the maize–bean intercrop and the fodder in the push–pull cropping system compensated for the loss.

This study shows that maize weevils and sap beetles have the potential to passively spread *Aspergillus* and *Fusarium* spores on pre-harvest maize. Spore loads varied between species, with weevils carrying more spores on their bodies than the other insects. The fact that maize weevils infested with mycotoxigenic fungi start infesting maize right in the field (before harvest) is a concern, because when the crop is harvested, there is a chance that the weevils will either spread to neighbouring fields or get into the farm stores. This cycle would perpetuate more ear rot in the field or the fungal contamination of stored grain, potentially increasing the levels of mycotoxins in grain for food and feed. It is recommended that further studies on plant–insect–mycotoxigenic fungi interactions are undertaken in the wake of climate change, which increases the abundance and diversity of pests.

## Figures and Tables

**Figure 1 insects-15-00995-f001:**
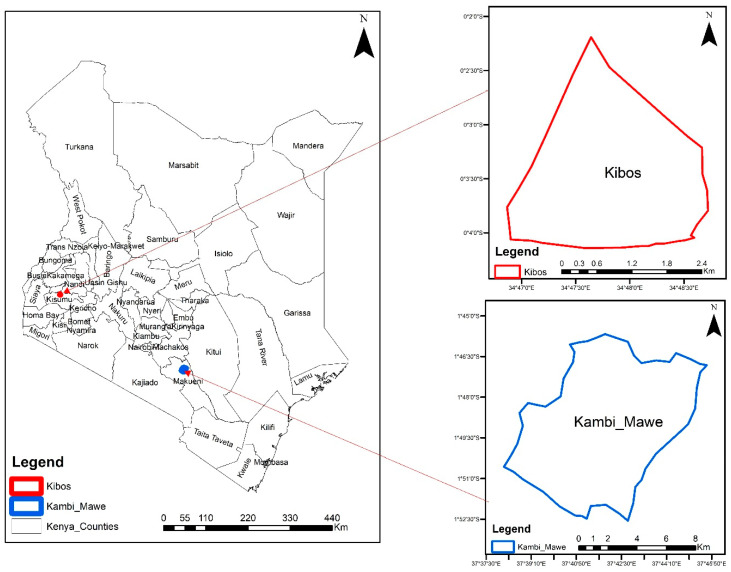
Location of experimental sites in Kibos, Kisumu County, and Kambi Mawe, Makueni County.

**Figure 2 insects-15-00995-f002:**
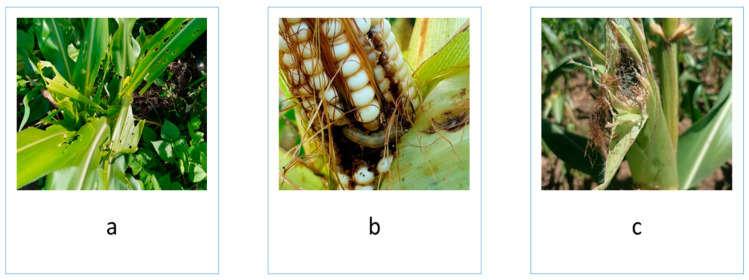
FAW damage on (**a**) maize foliage and (**b**) maize ears and (**c**) moulds on damaged maize ears.

**Figure 3 insects-15-00995-f003:**
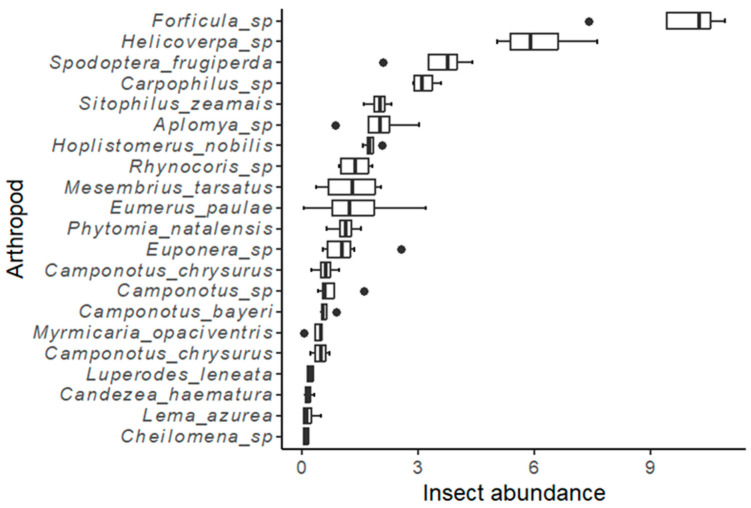
Whisker plots of the mean abundance of the most prevalent arthropod taxa (individuals/plant per sampling event) in the two seasons in Kisumu and Makueni. The dark line represents the median value, the whiskers represent the minimum and maximum values, and the dots represent the outliers.

**Table 1 insects-15-00995-t001:** Maize–legume intercrop varieties, spacing, and planting seasons for the on-station trials in Kibos, Kisumu County, and Kambi ya Mawe, Makueni County.

County	Maize Variety and Spacing (Inter-Row × Intra-Row)	Bean Variety	Year of Trial	Annual Rainfall
Kisumu	DK 8031 0.75 m × 0.3 m	GLP 2	Long season (March–August 2021) Short season (September–January 2021)	1714 mm
Makueni	DK 8031 0.9 m × 0.3 m	KAT B1	Long season (March–August 2021) Short season (September–January 2021)	828 mm

**Table 2 insects-15-00995-t002:** Percentage of FAW incidence (mean ± SE) on maize foliage and cobs and the severity of FAW damage to maize in different maize–legume intercropping systems (1 = low to 5 = high) in Kisumu and Makueni during the long and short rain seasons.

Location	Season	Cropping System	Percentage of FAW Damage Incidence on Foliage	FAW Severity on Foliage (1–5)	Percentage of FAW Damage Incidence on Cobs
Kisumu	Long rain	Maize monocrop	60.7 ± 2.6 a	3.1 ± 0.1	20.0 ± 5.8 a
		Maize–bean	47.0 ± 4.9 a	2.0 ± 0.1	20.0 ± 0.0 a
		Maize–bean–*Trichoderma*	43.7 ± 7.8 ab	2.0 ± 0.1	23.3 ± 6.7 a
		Push–pull	31.3 ± 5.0 b	1.2 ± 0.2	6.7 ± 6.7 a
		Mean *p* value	45.7 ± 3.9 0.030	2.08 ± 0.2 0.03	17.5 ± 3.1 0.22
	Short rain	Maize monocrop	78.3 ± 10.8 a	3.1 ± 0.1	40.0 ± 5.8 a
		Maize–bean	49.7 ± 4.5 b	2.1 ± 0.0	33.3 ± 3.3 ab
		Maize–bean–*Trichoderma*	41.7 ± 8.3 b	2.0 ± 0.0	23.3 ± 3.3 b
		Push–pull	45.0 ± 4.9 b	1.4 ± 0.1	26.7 ± 3.3 b
		Mean *p* value	53.7 ± 5.5 0.033	2.1 ± 0.2 0.03	30.8 ± 2.6 0.034
Makueni	Long rain	Maize monocrop	100.0 ± 0.0 a	3.8 ± 0.2	100.0 ± 0.0 a
		Maize–bean	100.0 ± 0.0 a	3.2 ± 0.2	100.0 ± 0.0 a
		Maize–bean–*Trichoderma*	100.0 ± 0.0 a	3.3 ± 0.3	100.0 ± 0.0 a
		Push–pull	43.3 ± 8.8 b	2.3 ± 0.2	56.7 ± 21.9 b
		Mean *p* value	85.8 ± 7.6 <0.001	3.14 ± 0.2 0.04	89.2 ± 7.3 <0.001
	Short rain	Maize monocrop	62.0 ± 15.3 a	3.4 ± 0.5	100.0 ± 0.0 a
		Maize–bean	56.7 ± 3.4 a	3.0 ± 0.0	100.0 ± 0.0 a
		Maize–bean–*Trichoderma*	42.3 ± 5.5 a	3.0 ± 0.0	100.0 ± 0.0 a
		Push–pull	42.7 ± 7.8 a	2.8 ± 0.2	78.7 ± 10.7 b
		*p* value	0.37	0.06	0.053

Means followed by the same letter within columns and individual treatments are not significantly different (Tukey’s honestly significant difference test, *p* ≤ 0.05).

**Table 3 insects-15-00995-t003:** Effects of site, season, and treatment on the mean ± SE fungal contamination (colony forming units = CFUs) of maize weevils (*Sitophilus zeamais*), sap beetles (*Carpophilus* spp.), earwigs (*Forficula* spp.), and sugar ants (*Camponotus* spp.) captured from Makueni and Kisumu.

Variable	Fungal Genera (CFUs/Insect)							
	*Aspergillus*				*Fusarium*			
	*Sitophilus zeamais*	*Carpophilus* spp.	*Forficula* spp.	*Camponotus* spp.	*Sitophilus zeamais*	*Carpophilus* spp.	*Forficula* spp.	*Camponotus* spp.
Site								
Makueni	116.5 ± 37.7 a	46.4 ± 19.0 a	36.9 ± 16.5 a	0.5 ± 0.5 b	47.3 ± 29.9 a	4.2 ± 4.2 a	0.4 ± 0.4 b	0.0 ± 0.0 a
Kisumu	15.8 ± 10.8 b	0.8 ± 0.8 b	14.6 ± 5.6 a	3.5 ± 1.5 a	60.5 ± 20.2 a	0.0 ± 0.0 a	43.3 ± 5.8 a	5.8 ± 5.8 a
Tukey’s HSD 0.05	66.8	33.3	33.9	2.7	93.5	6.03	30.9	11.1
Season								
Long rain	129.7 ± 37.5 a	45.5 ± 19.1 a	49.1 ± 16.3 a	2.6 ± 1.4 a	3.5 ± 1.4 b	0.0 ± 0.0 a	0.0 ± 0.0 b	0.0 ± 0.0 a
Short rain	2.6 ± 0.9 b	1.7 ± 1.1 b	2.3 ± 1.7 b	1.4 ± 0.8 a	104.4 ± 41.0 a	4.2 ± 3.0 a	43.8 ± 18.0 a	5.8 ± 5.8 a
Tukey’s HSD 0.05	66.9	33.3	33.9	2.7	93.5	6.03	30.87	11.9
Cropping system								
Sole maize	90.0 ± 58.7 a	24.2 ± 19.5 a	25.0 ± 22.7 a	5.8 ± 2.8 a	25.0 ± 17.8 a	5.3 ± 5.3 a	35.8 ± 24.3 a	0.0 ± 0.0 a
Maize–bean	89.3 ± 45.7 a	31.1 ± 28.1 a	33.8 ± 20.3 a	1.3 a ± 0.9 b	34.7 ± 31.7 a	0.0 ± 0.0 a	20.3 ± 19.2 a	0.0 ± 0.0 a
Maze–bean–T	40.0 ± 28.5 a	0.0 ± 0.0 a	31.3 ± 18.3 a	0.8 ± 0.8 ab	52.8 ± 39.5 a	3.1 ± 3.1 a	31.4 ± 21.5 a	11.7 ± 2.7 a
Push–pull	45.5 ± 28.1 a	39.2 ± 21.2 a	12.8 ± 6.0 a	0.0 ± 0.0 b	103.3 ± 69.3 a	0.0 ± 0.0 a	0.0 ± 0.0 a	0.0 ± 0.0 a
Tukey’s HSD 0.05	125.8	62.7	63.8	5.0	176.1	11.4	58.1	22.4

Means followed by the same letter within columns and individual treatments are not significantly different (Tukey’s honestly significant difference test, *p* ≤ 0.05). CFU—colony-forming units. N = 20—CFUs for 20 individuals per sample were determined. HSD = minimum difference considered significant between means.

**Table 4 insects-15-00995-t004:** Viable *Aspergillus* and *Fusarium* spores (CFUs/body part) from fecal matter and different body parts of earwigs, maize weevils, and sap beetles. The numbers in the brackets represent the percentage of individuals harbouring fungal spores on the respective body part. N = 20—spores for 20 individuals per taxon were determined.

Fungi	Insect	Number of *Aspergillus* and *Fusarium* Spores on Body Part (Mean ± SE)								% Total Infected Individuals
		Feces		Elytra		Head		Gut		
*Aspergillus*	*Forficula* spp.	0.33 ± 0.33	(1.65)	5.00 ± 2.89	(25.00)	0.67 ± 0.60	(3.35)	0.00 ± 0.00	(0.00)	26.00
	*S. zeamais*	0.00 ± 0.00	(0.00)	9.33 ± 3.28	(46.65)	6.67 ± 3.53	(33.35)	3.33 ± 1.76	(16.65)	52.50
	*C. dimidiatus*	1.33 ± 0.88	(6.65)	3.67 ± 2.03	(18.35)	5.33 ± 2.03	(26.65)	3.33 ± 2.03	(16.65)	27.10
	Mean	0.56 ± 0.34	(2.76)	6.00 ± 1.63	(30.00)	4.22 ± 1.50	(21.11)	2.22 ± 0.95	(11.10)	35.2
*F. verticilloides*	*Forficula* spp.	1.33 ± 1.33	(0.65)	2.67 ± 0.67	(13.35)	3.33 ± 0.88	(16.65)	0.67 ± 0.67	(3.35)	23.20
	*S. zeamais*	0.00 ± 0.00	(0.00)	5.33 ± 0.67	(26.65)	4.33 ± 0.88	(21.65)	0.33 ± 0.33	(1.65)	29.50
	*C. dimidiatus*	0.67 ± 0.67	(3.35)	5.33 ± 0.88	(26.65)	4.67 ± 0.33	(23.35)	2.67 ± 1.45	(13.35)	35.00
	Mean	0.67 ± 0.47	(1.33)	4.44 ± 0.58	(22.21)	4.11 ± 0.42	(20.55)	1.22 ± 0.60	(6.11)	29.23

The numbers in the brackets represent the percentage of individuals harbouring fungal spores on the respective body part. N = 20—spores for 20 individuals per taxon were determined.

**Table 5 insects-15-00995-t005:** Effects of site, season, and treatment on maize grain yield, grain spoilage, bean yield, *Desmodium* yield, and aflatoxin content.

Variable	Seasons	Treatment	Counties	
			Makueni	Kisumu
Maize yield (kg/ka)	Long rain season	Maize monocrop	2937.5 ± 406.4 a	7924.7 ± 196.4 a
		Maize–bean	2158.3 ± 61.7 a	6933.7 ± 346.1 a
		Maize–bean–*Trichoderma*	2639.2 ± 434.6 a	7322.3 ± 649.0 a
		Push–pull	2276.7 ± 64.3 a	7062.0 ± 575.2 a
	Short rain season	Maize monocrop	3396.0 ± 462.4 a	10,150.0 ± 1175.8 a
		Maize–bean	2520.0 ± 72.0 a	7516.7 ± 183.3 a
		Maize–bean–*Trichoderma*	3054.0 ± 276.9 a	8983.3 ± 799.1 a
		Push–pull	2644.0 ± 220.2 a	8433.3 ± 1322.0 a
Percentage of spoiled grain	Long rain season	Maize monocrop	41.3 ± 6.9 a	4.2 ± 1.6 ab
		Maize–bean	37.2 ± 1.7 ab	3.5 ± 1.7 b
		Maize–bean–*Trichoderma*	24.3 ± 2.5 ab	7.0 ± 1.3 a
		Push–pull	21.5 ± 5.8 b	2.2 ± 0.3 b
	Short rain season	Maize monocrop	37.3 ± 1.3 a	5.0 ± 2.9 a
		Maize–bean	29.7 ± 2.9 a	4.2 ± 0.8 a
		Maize–bean–*Trichoderma*	25.3 ± 5.0 a	0.8 ± 0.8 a
		Push–pull	26.0 ± 2.7 a	1.7 ± 0.8 a
100-kernel weight (g)	Long rain season	Maize monocrop	25.3 ± 1.3 a	32.9 ± 1.1 a
		Maize–bean	25.4 ± 1.1 a	29.8 ± 0.4 a
		Maize–bean–*Trichoderma*	28.2 ± 2.7 a	31.3 ± 1.2 a
		Push–pull	28.0 ± 2.8 a	29.9 ± 0.4 a
	Short rain season	Maize monocrop	27.4 ± 2.0 a	23.9 ± 2.2 a
		Maize–bean	26.3 ± 0.6 a	23.3 ± 1.7 a
		Maize–bean–*Trichoderma*	28.0 ± 1.5 a	25.6 ± 1.8 a
		Push–pull	27.2 ± 0.1 a	22.7 ± 2.2 a
Bean yield (kg/ha)	Long rain season	Maize monocrop	0.0 ± 0.0	0.0 ± 0.0
		Maize–bean	249.3 ± 20.4	388.5 ± 22.7
		Maize–bean–*Trichoderma*	239.3 ± 22.4	384.8 ± 27.9
		Push–pull	0.0 ± 0.0	0.0 ± 0.0
	Short rain season	Maize monocrop	0.0 ± 0.0	0.0 ± 0.0
		Maize–bean	122.0 ± 23.2	360.5 ± 14.5
		Maize–bean–*Trichoderma*	128.3 ± 2.0	322.7 ± 28.8
		Push–pull	0.0 ± 0.0	0.0 ± 0.0
Desmodium (kg/ha)	Long rain season	Maize monocrop	0.0 ± 0.0	0.0 ± 0.0
		Maize–bean	0.0 ± 0.0	0.0 ± 0.0
		Maize–bean–*Trichoderma*	0.0 ± 0.0	0.0 ± 0.0
		Push–pull	1983 ± 308.7	4991.7 ± 162.6
	Short rain season	Maize monocrop	0.0 ± 0.0	0.0 ± 0.0
		Maize–bean	0.0 ± 0.0	0.0 ± 0.0
		Maize–bean–*Trichoderma*	0.0 ± 0.0	0.0 ± 0.0
		Push–pull	916.7 ± 78.8	3675 ± 322.6
Aflatoxins (ppm)	Long rain season	Maize monocrop	10.6 ± 0.3 a	<LoD
		Maize–bean	10.4 ± 0.3 a	<LoD
		Maize–bean–*Trichoderma*	10.7 ± 0.4 a	<LoD
		Push–pull	6.6 ± 0.7 b	<LoD
	Short rain season	Maize monocrop	4.7 ± 0.1 ba	<LoD
		Maize–bean	3.9 ± 0.9 a	<LoD
		Maize–bean–*Trichoderma*	2.6 ± 0.5 a	<LoD
		Push–pull	3.0 ± 0.8 a	<LoD

LoD = limit of detection. Means followed by the same letters within a column and an individual treatment are not significant at *p* ≤ 0.05. Values in the table are means ± SE of the variables.

## Data Availability

The data supporting this study’s findings are available from the corresponding author upon written request. Voucher specimens were preserved at the Kenya Agricultural and Livestock Research Organization.

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
