# Peer review of "The Effect of Cropping Systems on the Dispersal of Mycotoxigenic Fungi by Insects in Pre-Harvest Maize in Kenya"

_insects, 2024, doi:10.3390/insects15120995_

Round 1
Reviewer 1 Report
Comments and Suggestions for Authors
Overall manuscript is written nicely. Introduction can be shortened a bit. Looks a bit lengthy. Objectives are well defined. Some parts in the methodology sections are needed to be clarify (i have given comments in this sections).
Results section need to be checked again statistically, at some points issues with SE is present like...same SE as MEAN, too high SE etc. I have given some comments in this section. Authors can go through these comments to improve this section.
Discussion section is written nicely, it would be better if author can add some recent references.
Authors are advised to go through all the comments mentioned in the reviewed file and revised MS accordingly.

Author Response
Ln 65 spelling of leafhopper checked and corrected ln 65
Ln 129 Word respectively added ln 138
Ln 141 insects were collected between 0800 and 1200 to ensure they were comparable. A change of timing would affect the numbers.
Ln 144 While collecting insects which stage or stages of insect were collected? OR focused was only on the adult collection?- adults and larva of Lepidoptera taxa.The adults and lepidopteran larvae were captured ln 152
Ramel, Gordon. 2020. “Key to the Orders of Insects:- Insect Identification Key.” Earth Life. April 20, 2020. https://earthlife.net/insect-identification-key/.LN 158
Lancashire, P.D.; H. Bleiholder; P. Langeluddecke; R. Stauss; T. van den Boom; E. Weber; A. Witzen-Berger. (1991). A uniform decimal code for growth stages of crops and weeds. Annals of Applied Biology, 119 (3): 561–601. Ln 202
Ln 236. Use symbol. The multiplication symbol was used ln 253
Ln 240 input the author of BBCH scale input ln 150
CHANGED TO FAW ln 163, 264, 265,275,278,280,282,284, 438, 446
Ln 254-255 use only mean without SE . corrected Ln271-272
Table 2 . P values missing . the p values added

Reviewer 2 Report
Comments and Suggestions for Authors
In the reported study the authors have investigated the effects of different cropping systems on the abundance of pest insects and mycotoxigenic fungi as well as mycotoxin contamination in maize. Experiments were performed in a randomized complete block design with three replicates plots at two locations in Kenya. Spodoptera frugiperda turned out to be the most damaging pest. Intercropping and push – pull approaches resulted in decreased damage caused by S. frugiperda. Mycotoxigenic fungi including Aspergillus spp. and Fusarium verticillioides were detected on different insect pests present, which suggests a possible role of these insects in dispersal of the fungi in pre-harvest maize.
The authors present a large and interesting data set in this manuscript. A variety of very interesting factors have been assessed and results of the study support the importance of intercropping or push-pull approaches to reduce pest insect abundance, yield loss and mycotoxin contamination. Approaches taken to address the research question are appropriate. However, the manuscript in its current stage lacks details and precision throughout. Particularly the sample collection is poorly described, which makes it difficult to follow how and where exactly samples have been collected. Therefore, it is difficult to assess accuracy of the experimental procedures. Authors should be careful with the interpretation of their results. For instance, no experiments were presented, which give proof that insects truly disperse mycotoxigenic fungi. The data provided only suggest a dispersal effect because fungal inoculum has been isolated from the insects.
Specific comments:
L36 modify: maize-bean intercrop with “additional Trichoderma harzianum application”
L34 “to spread Aspergillus spores” – what about other species (L40)?
L61/62 Which are these theories? References?
L63 “Intercropping has been associated…” this sentence is a repetition of the sentence in line 60
L78 T harzianum does not combat aflatoxins. It inhibits or controls aflatoxin producers, it helps to reduce aflatoxin contamination.
L101 I am not sure how unique the approach is. Maybe rather “the study was focused on..)
L118 how and when was Trichoderma sprayed?
L124 how was the RCBD organized, was the lay out the same at both sites? Possibly a figure as supplemental material? Were there any buffer spaces between the plots?
L140 how were the arthropods captured? From each plot and treatment? From x plants per plot? Throughout the plot?
L151 “Since the FAW…” this sentence does not make sense to me. In addition, according to figure 3 Helicoverpa was more prominent…
L153 on how many plants was incidence on foliage determined? In each plot? X plants per plot? Where in the plot?
L159 Where were the twenty individuals caught? the same comment as L140. How thorough and reproducible is the procedure? I assume quite a bit of the spores remain attached to the insects…. Are there any controls?
L165 plated not placed
L172 how can you test the mode of spread with this analysis? This analysis only tests for presence of spores…
L173 Again how and where were the insects collected? In or outside of the experimental field? Why only at one site?
L189 how were the five ears selected? From each plot? What do you mean with “batch”?
L208 1m2 areas per plot?
L279 Examples of other fungal taxa were observed?
L293 how were the fungal species identified?
L316 results on Desmodium are not represented in the text
L344 there should be a reference for the statement in the first sentence
L392 what does “heavy” mean?
L426 how was the range of damage reduction in the cited studies? Was it comparable to the results reported here?
L429 here you argue with drought but in L392 you mention long and heavy rain… this is contradictory. Also, L440 drought stress at flowering stage is currently not reported in the manuscript. How exactly was the season at the two sites? Typically, fungal diseases and contamination is more severe in humid conditions…
L444 where are the data that show the general insect abundance in the different cropping systems?
L447 “quickly”?
L448 The study only documents the potential for spread by these insects.
Table 2
Footnote: withing columns and individual treatments…
Table 3
Footnote: withing columns and individual treatments…
Tukey HSD 0.05, what are the numbers we see? Provide information in the footnote.
Table 4
Makueni only? If yes mention in headings. I assume the numbers represent CFUs? This should be mentioned.
Table 5
Footnote: “withing columns “and individual” treatments…
Author Response
Specific areas
However, the manuscript in its current stage lacks details and precision throughout. Particularly the sample collection is poorly described, which makes it difficult to follow how and where exactly samples have been collected. Therefore, it is difficult to assess accuracy of the experimental procedures. Authors should be careful with the interpretation of their results. For instance, no experiments were presented, which give proof that insects truly disperse mycotoxigenic fungi. The data provided only suggest a dispersal effect because fungal inoculum has been isolated from the insects.
We agreed with the comment and have improved the precision of the manuscript as guided by the reviewer.
Specific comments:
L36 modify maize-bean intercrop with “additional Trichoderma harzianum application”-the words additional added- Ln 36
L34 “to spread Aspergillus spores” – what about other species (L40)?- edited to read mycotoxigenic fungi, we agree that not only Aspergillus was isolated -Ln 295 we mention other fungi isolated over and above the Aspergillus and Fusarium
L61/62 Which are these theories? References?-theories and 3 references added, ln 61-70
L63 “Intercropping has been associated…” this sentence is a repetition of the sentence in line 60-edited to remove repetition (the sentence was deleted)
L78 T harzianum does not combat aflatoxins. It inhibits or controls aflatoxin producers, it helps to reduce aflatoxin contamination (we agreed and edited to combating soil borne fungi at line 82-83
L101 I am not sure how unique the approach is. Maybe rather “the study was focused on..)-we agreed with the reviewer and Agreed and edited to focused on 106
L118 how and when was Trichoderma sprayed? (Trichoderma was sprayed in the planting holes)Trichoderma harzianum was mixed in water at 1:5, product to water ratio as per manufacturer's instructions, and sprayed on the maize planting holes ln 127-129
L124 how was the RCBD organized, was the lay out the same at both sites? Possibly a figure as supplemental material? Were there any buffer spaces between the plots? (a standard design used provided as supplemet. Only a 2 metres path separated the plots. Cross treatment effect was minimized by using large plots 30mx30m and only picking insects and the sample from the 5 innermost rows. Ln 132-134 and supplementary Figure added
L140 how were the arthropods captured? From each plot and treatment? From x plants per plot? Throughout the plot? (hand picking or by use of a pooter from 20 pre-tagged plants to minimize disturbance Ln 148-153
L151 “Since the FAW…” this sentence does not make sense to me. In addition, according to figure 3 Helicoverpa was more prominent… The most damaging was FAW but the larvae numbers of the H. zea were more. Ln 162
L153 on how many plants was incidence on foliage determined? In each plot? X plants per plot? Where in the plot?(20m plants added) from 20 plants-ln 165
L159 Where were the twenty individuals caught? the same comment as L140. How thorough and reproducible is the procedure? I assume quite a bit of the spores remain attached to the insects…. Are there any controls? ((all the insects found on the ears and silks of the 20 pre-tagged plants. Per plot were captured).In case they were more than 20, a systematic sampling was done depending on the total number of insects per insect per plot collected. Ln 150-153
L165 plated not placed= Agreed and collected placed replaced with plated
L172 how can you test the mode of spread with this analysis? This analysis only tests for the presence of spores… A confirmation was done in another study in a greenhouse (data not shown). On this one it was just the ability/ possibility implied either as active or passive spread )
L173 Again how and where were the insects collected? In or outside of the experimental field? Why only at one site? (all the insects found on the ears and silks of the 20 pre-tagged plants. Per plot. The 20 pre-tagged plants were in the 5 innermost rows of the plot.ln 171-172
L189 how were the five ears selected? From each plot? What do you mean with “batch”?- systematically (every fourth plant from the 20 pre-tagged per plot. The word batch deleted ln 204-206
L208 1m2 areas per plot? 4 replicates of 1m2 per plot then averaged Ln 224
L279 Examples of other fungal taxa were observed?-Penicillium and Diplopodia added ln 296
L293 how were the fungal species identified? Based on colony morphology using keys Klich, 2002 for Aspergillus and Summerel 2006 for Fusarium and confirmed by molecular techniques (data not shown) ln 201-202
L316 results on Desmodium are not represented in the text-inserted ln 343-344
L344 there should be a reference for the statement in the first sentence-inserted Njeru et al, 2019 ln 362-363
L392 what does “heavy” mean? -over 20mm of rainfall per day. Ln 411
L426 how was the range of damage reduction in the cited studies? Was it comparable to the results reported here?(comparable results in this study showed that the incidence between maize monocrop and push-pull reduced by 67 while in the cited papers, it reduced by 43-47. 444-446
L429 here you argue with drought but in L392 you mention long and heavy rain… this is contradictory. Also, L440 drought stress at flowering stage is currently not reported in the manuscript. How exactly was the season at the two sites? Typically, fungal diseases and contamination is more severe in humid conditions… The heavy rainfall was in Kisumu with an annual rainfall of 1714mm as compared to Makueni 828 mm. the two sites had very different weather patterns especially rainfall ln 417-418
L444 where are the data that show the general insect abundance in the different cropping systems? (under review in another paper not included here
L447 “quickly”?-deleted
L448 The study only documents the potential for spread by these insects ;word can added to show possibility ln 467. Though a confirmation was done and published in another article
Table 2
Footnote: withing columns and individual treatments… corrected to within(added) ln 286
Table 3
Footnote: withing columns and individual treatments…(added) ln 325
Tukey HSD 0.05, what are the numbers we see? Provide information in the footnote.-HSD explained as Minimum difference considered significant between means.
Table 4
Makueni only? If yes mention in headings. I assume the numbers represent CFUs? This should be mentioned.=added ln 330
Table 5
Footnote: “within columns “and individual” treatments…(added) Ln 355

Round 2
Reviewer 2 Report
Comments and Suggestions for Authors
I have reviewed the manuscript previously and therefore, have focused this review on the issues raised previously.
The others have addressed all the issues raised. However, in some cases the modifications are not satisfactory. I wished the authors would have been more careful, it looks like the correction have been done in a hurry.
L63 rather “conclusion” not “key”?
L66 “secondly” but where is “firstly”? I think this sentence is rather an explanation of the previous statement.
L134 “supplementary fig.1” in parentheses?
L163 “Incidence” not with capital letter
L164 now it reads “The (FAW) were identified”… The FAW larvae were… then modify next sentence.
L295 better write “other fungal genera detected included Penicillium…
L309 comment in reply to review is not satisfactory. Identification with molecular techniques has to be specified. Which marker gene?
L329 (CFU/ body part) not (DFU/part)
L333 “fungal spores on the respective” not “at the”
L411 The rain season was “high” – does not make sense
L468 beetles have the potential to passively spread” not “can”. If this has been shown in another study as stated in the reply, the publication has to be mentioned.
Author Response
I have reviewed the manuscript previously and therefore, have focused this review on the issues raised previously.
The others have addressed all the issues raised. However, in some cases the modifications are not satisfactory. I wished the authors would have been more careful, it looks like the correction have been done in a hurry.
L63 rather “conclusion” not “key”? I have re-written paragraph ln 60-68 to show the various explanations on the increase in entomofauna in intercrop systems rather than show them as separate theories
L66 “secondly” but where is “firstly”? I think this sentence is rather an explanation of the previous statement : The paragraph has been re-written to shown a continuation of the explanation ln 60-68
L134 “supplementary fig.1” in parentheses? Parentheses added as suggested ln132
L163 “Incidence” not with capital letter (word incidence corrected ln 164)
L164 now it reads “The (FAW) were identified”… The FAW larvae were… then modify next sentence. The sentence corrected to read ''The FAW larvae were noted as the most destructive herbivores damaging both the foliage and the cobs across the two regions, and since records of effective management using the push-pull method have been reported, data on the incidence and severity of the attack was determined on 20 plants per plot randomly selected from the 5 innermost rows'' ln 160-164
L295 better write “other fungal genera detected included Penicillium … Agreed with review and edited as suggested 293-295
L309 comment in reply to review is not satisfactory. Identification with molecular techniques has to be specified. Which marker gene Agreed with the reviewer and the markers used were mentioned. The data is in another publication currently in preparation. The calmodulin and the elongation factor gene markers were used to identify the Aspergillus and Fusarium isolates respectively [Riungu et al., in prep.].
L329 (CFU/ body part) not (DFU/part) corrected to CFU/ body part as suggested by the reviewer
L333 “fungal spores on the respective” not “at the” agreed and corrected to fungal spores on the respective……………………
L411 The rain season was “high” – does not make sense.. Agreed and corrected to read The rains during the long rain cropping season were high (over 20 mm per day on some days),
L468 beetles have the potential to passively spread” not “can”. If this has been shown in another study as stated in the reply, the publication has to be mentioned. Agreed to remain with the aspect of only found harbouring the spores so changed to beetles have the potential to passively spread as suggested by reviewer.
